# Evolution of Microstructure, Mechanical Properties, and Corrosion Resistance of Mg–2.2Gd–2.2Zn–0.2Ca (wt%) Alloy by Extrusion at Various Temperatures

**DOI:** 10.3390/ma16083075

**Published:** 2023-04-13

**Authors:** Hüseyin Zengin, Soner Ari, Muhammet Emre Turan, Achim Walter Hassel

**Affiliations:** 1Institute of Chemical Technology of Inorganic Materials (TIM), Johannes Kepler University Linz, 4040 Linz, Austria; 2Valfsel Armatür Sanayi A.S., 45030 Manisa, Türkiye; 3Iron & Steel Institute, Karabuk University, 78100 Karabük, Türkiye

**Keywords:** magnesium, homogenization, extrusion, dynamic recrystallization, tensile properties, corrosion

## Abstract

The current investigation involved casting the Mg–2.2Gd–2.2Zn–0.2Ca (wt%) alloy (GZX220) through permanent mold casting, followed by homogenization at 400 °C for 24 h and extrusion at 250 °C, 300 °C, 350 °C, and 400 °C. Microstructure investigations revealed that α-Mg, Mg–Gd, and Mg–Gd–Zn intermetallic phases were present in the as-cast alloy. Following the homogenization treatment, a majority of these intermetallic particles underwent partial dissolution into the matrix phase. α-Mg grains exhibited a considerable refinement by extrusion due to dynamic recrystallization (DRX). At low extrusion temperatures, higher basal texture intensities were observed. The mechanical properties were remarkably enhanced after the extrusion process. However, a consistent decline in strength was observed with the rise in extrusion temperature. The corrosion performance of the as-cast GZX220 alloy was reduced by homogenization because of the lack of corrosion barrier effect of secondary phases. A significant enhancement of corrosion resistance was achieved by the extrusion process.

## 1. Introduction

Magnesium holds the distinction of being the lightest structural metal [1]. As a point of comparison, it is 35% less dense than aluminum (2.7 g cm^−3^), which is among the most commonly used lightweight metals [2,3]. In addition, it has recently gained more attention thanks to its exceptional castability, formability, and specific strength properties. These qualities make it a promising option for various engineering applications [4].

Magnesium exhibits limited formability at low temperatures owing to its hexagonal crystal structure (HCP) with limited slip planes at room temperature [5]. Typically, hot forming techniques are mostly conducted between 300 °C and 450 °C for magnesium alloys, where more slip planes are active [5,6]. Elevated temperatures tend to increase extrusion speeds while reducing extrusion pressures. However, the maximum attainable temperature varies with the alloy, as materials with higher alloy content possess lower solidification temperatures, rendering them more vulnerable to initial melting and hot cracking [7]. Furthermore, higher extrusion temperatures can result in excessive heating of the extrudates, causing rapid grain growth and reduced strength of the final product [8]. The extrusion temperature affects the recrystallization mechanism and also enables the determination of the strength.

There is consensus that alloying is among the most prevalent techniques to improve the characteristics of magnesium. Zinc (Zn), which is extensively employed in magnesium alloying, increases strength through several hardening mechanisms [9,10]. In addition, Zn has the ability to boost the corrosion performance of magnesium by assisting in the development of a protective layer of oxide film [11]. Moreover, the ability of rare earth elements to improve the strength of magnesium at high temperatures and to resist creep is widely acknowledged [12,13,14]. Several studies concentrated on the role of gadolinium (Gd) in the characteristics of magnesium alloys [15,16,17,18,19]. According to Stanford et al. [15], minor quantities of Gd gave rise to a notable refinement of recrystallized grains, and recrystallization was strongly impeded by Gd additions at higher amounts. A study by Pourbahari et al. [17] focused on the impact of the ratio of Gd to Al on magnesium alloys and showed that the optimal combination of mechanical properties was achieved at a ratio of 1. Shi et al. [18] showed that the increasing Gd addition to Mg–Zn–Y alloys deteriorated the corrosion resistance because of the presence of micro-galvanic sites around Mg–Zn–Gd particles, whereas the compressive test results revealed an improvement of strength and ductility by Gd additions. A recent research by Gu et al. [20] demonstrated that adding up to 4 wt% of Gd to magnesium effectively enhanced its basal slip resistance compared to prismatic slip resistance, ultimately leading to an increase in the ductility of the magnesium. Furthermore, another study suggested that the presence of Gd, particularly when combined with Zn, may influence the texture development of magnesium and subsequently enhance its mechanical strength by increased bonding of solute elements to grain boundaries [21]. A parameter to be considered while designing magnesium alloys is the “electronic work function (EWF)”, which provides insight into the behavior of electrons in the metal. According to Liu and Li [22], if the EWF value of the dissolved element is smaller than that of Mg (3.66 eV), both strength and ductility can increase. Strength increase occurs with solid-solution strengthening. The ductilizing effect is due to the dissolved element creating cross-slip stress softening (*a*-type dislocations) that provides dynamic recovery during deformation and new nucleation zones for pyramidal <*c* + *a*> dislocations as a result of lower I_1_ stacking fault energy. Zn, with an EWF value of 4.33 eV, can effectively enhance the strength of magnesium through solid-solution strengthening. Gd has an EWF value of 3.17 eV and is an effective alloying element in increasing both the strength and ductility of magnesium. Another element that can have a similar effect is calcium (Ca), which is frequently used in magnesium alloys with an EWF of 2.87 eV. It was reported that alloying Mg–Zn alloy with Ca can give rise to a decline in the intensity of the basal texture, a grain size reduction, and a substantial enhancement in the tensile elongation [23]. It was also shown that the minor Ca addition to a Mg–Zn–RE–Zr alloy reduced the yield asymmetry owing to the smaller grains and weaker texture [24].

Due to the promising results from adding Zn, Gd, or Ca to magnesium alloys, the alloy containing 2.2 wt% Zn, 2.2 wt% Gd, and 0.2 wt% Ca was designed and fabricated by casting, homogenization, and extrusion at different temperatures in the current study. The goal is to develop a magnesium alloy that exhibits ideal corrosion resistance, strength, and ductility while also exploring the correlation between microstructure and corrosion.

## 2. Experimental Procedure

The studied alloy, GZX220, was produced via permanent mold casting after melting in an induction furnace. First, melting of pure Mg ingots (>99.9%) was completed at 750 °C. Following this, Mg–25Gd and Mg–25Ca master alloys and pure Zn chips (>99.9%) were added to the molten mixture. To provide a homogenous composition, a mechanical stirring for 20 min was applied to the mixture. After that, the temperature was lowered to 720 °C, and the molten metal was immediately poured into a cylindrical steel mold preheated at 200 °C. The final cast product had dimensions of 42 mm in diameter and 320 mm in length. During the melting and pouring processes, a shielding gas mixture (CO_2_ + 1% SF_6_) was continuously supplied. The chemical composition analysis was conducted by wavelength dispersive X-ray fluorescence (XRF), and the results are given in Table 1. Following casting, the resulting parts were homogenized at 400 °C for a duration of 24 h and water quenched. Then, the samples were subjected to machining to produce cylindrical bars for extrusion. The extrusion process was performed using four distinct temperatures: 250 °C, 300 °C, 350 °C, and 400 °C. Prior to each extrusion process, the dies were preheated till they reached the desired temperature. Subsequently, the samples were extruded utilizing an extrusion ratio of 10:1 and a ram speed of 0.3 mm s^−1^, after which they were air-cooled.

Optical (OM) and scanning electron microscopies (SEM) were utilized to perform microstructural characterizations. Samples obtained from the GZX220 alloy were ground and polished. A picral solution was used for etching. The Rigaku Ultima IV X-ray diffractometer (XRD) was employed to analyze the constituent phases at a scanning speed of 2° min^−1^ and diffraction angles ranging from 20° to 80°. The experimental (0002), (10$\bar{1}$0), and (11$\bar{2}$0) pole figures were also measured by the XRD method using Cu Kα radiation. The data were obtained on 5° tilt steps from 15° to 90° and azimuthal rotations over the entire 360° range. The thermal stabilities of the constituent phases were analyzed by differential scanning calorimetry (DSC) using the HITACHI DSC7000 series instrument with a scanning rate of 10 °C min^−1^. The tensile test specimens had a 5 mm diameter and a 25 mm gauge length. The tests were performed in a universal test machine (Zwick/Roell Z600) at a strain rate of 10^−3^ s^−1^. Each test was repeated a minimum of three times.

The corrosion behaviors of the samples were characterized by constant immersion and electrochemical corrosion tests in 3.5 wt% NaCl solution at room temperature. For immersion corrosion tests, the cylindrical samples were immersed in the chloride solution for up to 48 h. A solution of chromic acid was employed for the removal of loosely bound products on the surface. The electrochemical corrosion tests were performed by CompactStat potentiostat (Ivium Technologies, Eindhoven, The Netherlands) after immersing in the chloride solution and monitoring the open-circuit potential (*E_ocp_*) values for 30 min. A classical three-electrode setup, which included a platinum wire serving as the counter electrode, an Ag|AgCl|3 M KCl as the reference electrode, and the test specimen as the working electrode, was used. The polarization curves were generated at a scan rate of 1 mV s^−1^ within the voltage range of −0.5 to +0.5 V relative to *E_ocp_*. The cathodic corrosion current density (*i_corr_*) values are estimated from the point where the linear fit taken starting from approximately 100 mV below the corrosion potential (*E_corr_*) intersects with the horizontal line drawn from *E_corr_*. The frequency range used for EIS was from 100 kHz to 0.1 Hz with a sinusoidal voltage amplitude of 10 mV.

## 3. Results and Discussion

### 3.1. Microstructural Characterizations

#### 3.1.1. Microstructure of the As-Cast and Homogenized Alloys

Figure 1 demonstrates the XRD patterns of the as-cast and homogenized GZX220 alloy. The as-cast alloy displayed peaks that matched with α-Mg, Mg_3_Gd_2_Zn_3_, Mg_3_Zn_6_Gd, Ca_2_Mg_6_Zn_3_, and Mg_5_Gd phases. The formation of three forms of ternary phases is possible in the Mg–Zn–Gd system with the specific type depending on the composition of Zn and Gd. These ternary phases are the W-phase (Mg_3_Zn_3_Gd_2_), I-phase (Mg_3_Zn_6_Gd), and X-phase (Mg_12_ZnGd) [25,26,27]. The W-phase is the most prevalent phase among them, occurring when the ratio of Zn to Gd is below 6 [28]. Furthermore, it was reported that when the atomic ratio of Zn to Gd was in the range from 1.5 to 40, the I-phase could form together with the W-phase [29]. In the present study, this ratio was around 2.4. Thus, no considerable amount of I-phase should be expected to form. In Figure 1, a low-intensity peak belonging to the I-phase was observed. After the homogenization treatment, the peaks for the Ca_2_Mg_6_Zn_3_ phase showed a significant reduction, whereas the peaks for the W-phase and the I-phase remained nearly the same. The transition temperatures of the phases in the as-cast alloy that were analyzed by DSC measurement are illustrated in Figure 2. Three peaks indicating endothermic reactions were detected at temperatures of 406 °C, 503 °C, and 628 °C. Considering that the Ca_2_Mg_6_Zn_3_ phase displayed a significant amount of dissolution during homogenization at 400 °C, it can be deduced that the transition peak at 406 °C corresponded to Ca_2_Mg_6_Zn_3_. Furthermore, the W-phase exhibited a transition at 503 °C, and the peak at 628 °C indicated the melting of the alloy. Similar thermal properties of the constituent phases were reported in previous studies [26,30,31]. On the other hand, the DSC results did not reveal any peaks associated with the I-phase and Mg_5_Gd, as their fractions were expected to be very low according to the XRD results, which showed very weak intensities for these phases.

The optical microstructures and the grain size distribution histograms of the as-cast and homogenized GZX220 alloy are displayed in Figure 3. The as-cast alloy consisted of α-Mg grains and intermetallic particles, which formed somewhat a partially connected network throughout the microstructure. Upon homogenization treatment at 400 °C, some of these particles dissolved partially into the matrix phase.

The SEM micrographs of the GZX220 alloy in both as-cast and homogenized conditions are depicted in Figure 4. The energy-dispersive X-ray spectrometer (EDS) analyses of the locations marked in the micrographs are provided in Table 2. The second-phase particles in the as-cast alloy exhibited several morphologies such as globular, strip-like, and lamella-like. For example, the phase indicated as 1 had eutectic lamella-like structures. EDS analysis indicated that this phase was enriched with Mg, Zn, and Gd, with a similar atomic ratio to that of Mg_3_Gd_2_Zn_3_. The globular second phase marked with 2 contained similar amounts of Mg, Zn, and Gd. Thus, that globular phase can also be identified as Mg_3_Gd_2_Zn_3_. On the other hand, the short rod-like phase indicated by 3 was mainly composed of Mg, Zn, and Ca elements. Their atomic ratio was very similar to that of the Ca_2_Mg_6_Zn_3_ phase, indicating that the Ca_2_Mg_6_Zn_3_ phases were mainly present as discrete short rod-like particles with low fractions. The strip-like phase marked with 4 contained high concentrations of Mg together with lower amounts of Zn and Gd. This phase was compatible with neither Mg_3_Gd_2_Zn_3_ nor Mg_3_Zn_6_Gd. It was thought that the high amount of Mg in this phase was likely due to the low thickness of the particle, which allowed the electron beam to penetrate through the matrix phase. Following the homogenization process, the volume fraction of the second phases showed a reduction from 5.1% to 3.3%. The morphologies of the second phases also displayed an alteration. For example, the lamella-like eutectic structure in the as-cast alloy was transformed to a skeleton-like morphology, and the thick strip-like particles became either thin rods or discrete tiny particles due to the partial dissolution. During the EDS analyses of the homogenized alloy, no evidence of the Ca_2_Mg_6_Zn_3_ phase was found, which confirmed the results of XRD and DSC. The points marked with 1, 3, and 4 in the homogenized alloy showed similar compositions of Mg, Zn, and Gd with the Mg_3_Gd_2_Zn_3_ phases detected in the as-cast state. Several earlier studies also reported that the W-phase did not dissolve during the homogenization of Mg–Zn–Gd alloys [27,32,33]. Dissimilarly, the tiny particle with sharp corners marked with 2 in the homogenized alloy was rich in only Mg and Gd, implying the formation of the Mg–Gd binary phase. It is inferred that this phase corresponded to Mg_5_Gd due to the comparable atomic ratio of Mg to Gd. Similar formations of Mg_5_Gd in Mg–Zn–Gd systems were reported elsewhere [17,25].

#### 3.1.2. Microstructure of the Extruded Alloys

Figure 5 demonstrates the microstructures obtained by extrusion at various temperatures. In comparison to the conditions prior to extrusion, a considerable reduction in grain size due to the dynamic recrystallization (DRX) and a fragmentation of intermetallic particles were achieved by extrusion at all the extrusion temperatures. However, it seemed that the extrusion temperature noticeably influenced the degree of DRX, in which a grain structure that had two distinct modes comprised of fine DRXed grains and large non-DRXed grains in the alloy extruded at low extrusion temperatures was replaced with the fully DRXed unimodal fine grain structure as the extrusion temperatures increased up to 400 °C. Evidently, increasing extrusion temperature gave rise to an increase in the DRXed grain size but also improved the degree of DRX. The alloy extruded at 250 °C exhibited very fine equiaxed DRXed grains (~2.1 ± 0.1 µm), and non-DRXed grains were subjected to deformation. The average size of the DRXed grains showed a gradual increase to 5.2 ± 0.4 µm, 6.5 ± 0.5 µm and 7.4 ± 0.5 µm as the extrusion temperature further rose to 300 °C, 350 °C and 400 °C, respectively. The impact of temperature on the DRX mechanism was the subject of several studies [34,35,36]. Shahzad and Wagner [34] showed that the fine DRXed grains at low extrusion temperatures were because of the formation of new crystals at the twin boundaries, whereas at elevated extrusion temperatures, the initial grain boundaries were the nucleation sites for new crystals. Since DRX is a thermally activated metallurgical phenomenon, increasing extrusion temperature favored the DRX. However, excessive processing temperatures may lead to undesirable grain coarsening due to accelerated atomic diffusion [36]. Hence, the largest grain size after undergoing DRX was found for the alloy extruded at 400 °C.

Figure 6 presents the SEM micrographs of the GZX220 alloy extruded at various temperatures. The broken second-phase particles can be clearly seen (white regions) in the figures. Upon increasing extrusion temperatures, the second-phase particles became finer and oriented themselves along the extrusion direction more regularly. The probable cause of this was the partial dissolution of these particles during extrusion at high temperatures, in which a further increase in initial extrusion temperature would occur by additional heat generation [8,37]. Furthermore, the majority of the DRXed grains formed preferentially in the vicinity of the fine second-phase particles, especially in the alloys extruded at 250 °C and 300 °C. The second-phase particles can affect the grain growth during DRX depending on the particle size by either promoting DRX through a phenomenon called particle-stimulated nucleation (PSN) or impeding the grain growth and accordingly retarding the DRX by the Zener pinning effect [38,39]. The former occurs as the diameter of the particle is greater than 1 µm, and the latter occurs when particles smaller than 1 µm are present in the microstructure. The majority of the broken secondary phases in Figure 6 were greater than 1 μm, and thus, PSN was the main mechanism in the extruded alloys. However, the efficiency of the PSN seemed to cease with the rise in the extrusion temperature. The correlation between PSN and the Zener–Hollomon parameter (Z), which depends on the strain rate and deformation temperature, can explain this phenomenon [38,40]. As the Z parameter declines, which occurs at higher extrusion temperatures with constant strain rates, the likelihood of the PSN mechanism to occur also decreases. Therefore, the efficiency of PSN showed a decrease as the extrusion temperature increased.

Figure 7 illustrates the (0002), (10$\bar{1}$0), and (11$\bar{2}$0) pole figures of the homogenized and extruded GZX220 alloys. The intensities of the (0002), (10$\bar{1}$0), and (11$\bar{2}$0) planes were measured for the homogenized alloy, yielding maximum values of 7.9, 6.6, and 4.2 multiples of random distribution, respectively. The pole figures of the homogenized alloy demonstrated that the unit cells lacked any preferred crystallographic orientation and were instead randomly oriented. However, the extruded alloys produced strong basal textures, in which (0002) basal planes with <10$\bar{1}$0> directions of the majority of the grains were aligned with the extrusion direction. The basal texture intensities were measured at 12, 8.2, 6.5, and 5.8 for the alloys extruded at 250 °C, 300 °C, 350 °C, and 400 °C, respectively. The decreased basal pole intensities with increasing extrusion temperature were primarily related to the degree of DRX. Several studies reported that the deformed parent non-DRXed grains exhibited much higher basal texture intensities than the fine DRXed grains [41,42,43]. Thus, the alloys with higher volume fractions of DRXed grains expectedly showed texture weakening and randomization. Furthermore, the intensity of the (10$\bar{1}$0) pole decreased as the extrusion temperature increased from 250 °C to 300 °C. A further increase in the temperature did not lead to a considerable change. There was no significant variation in the (11$\bar{2}$0) pole intensities among all samples.

### 3.2. Mechanical Properties

The engineering stress–strain curves corresponding to the tensile tests conducted until failure at room temperature are demonstrated in Figure 8. Table 3 presents the yield (YS) and ultimate tensile strengths (UTS) and strain at failure values derived from Figure 8. The as-cast alloy displayed moderate mechanical properties with a YS of 114 ± 6 MPa, a UTS of 200 ± 7 MPa, and an elongation to fracture of 8.3 ± 1.4%. After the homogenization treatment all the tensile properties deteriorated by around 20%. That is to say, the mechanical properties were negatively influenced by the microstructural changes brought about by the homogenization treatment. As mentioned earlier, homogenization treatment left some clusters of non-homogeneously distributed, skeleton-like, and sharp-edged second-phase particles behind. Hence, the formation of premature cracks around these particles might have occurred, and accordingly, the ductility and strength were considerably reduced.

The mechanical properties showed a remarkable improvement by the extrusion process. The alloy extruded at 250 °C displayed the highest strength with a TYS of 321 ± 4 MPa and a UTS of 326 ± 4 MPa, while the value of strain at failure can be considered as low (8.8 ± 1.1%). The strength of the alloys continually decreased with increasing extrusion temperatures. On the other hand, the strain at failure values increased up to the extrusion temperature of 350 °C, above which it decreased. The enhanced mechanical properties through the extrusion process can be attributed to several mechanisms. As illustrated in Figure 5, the very fine structure of DRXed grains was achieved after the extrusion process allied with the distribution of broken second-phase particles. Moreover, the degree of DRX and the grain size produced through DRX were significantly influenced by the extrusion temperature. Thus, it can be inferred that the alterations in the mechanical properties due to changes in extrusion temperature can be mainly attributed to DRX mechanisms present in the extruded alloys. According to the Hall–Petch theory [44,45], an improvement of yield strength can be achieved by grain refining since an increased amount of grain boundaries hinders dislocation movement during plastic deformation. DRXed grains in the alloy, which was extruded at 250 °C, had an average size of 2.1 ± 0.1 µm. However, the superior strength values cannot only be attributed to such a small DRXed grain size since a full DRX was not achieved at that extrusion temperature. This indicated that the non-DRXed grains would affect the strength improvement more efficiently than the DRXed grains. The non-DRXed grains were exposed to deformation during the extrusion procedure, causing them to elongate along the direction of extrusion. Thus, a strong work hardening occurred within these grains, leading to a generation of a higher number of dislocations [35,46]. Furthermore, during the plastic deformation, {10$\bar{1}$1}–{10$\bar{1}$2} double twinning may serve as sites for crack initiation and can decrease the mechanical properties of the alloy [47]. It was reported that a larger amount of double twinning was observed in non-DRXed grains compared to those in DRXed grains [37,48]. Such extension twinning formation within the non-DRXed grains can be seen in the microstructures of the alloys extruded at 250 °C and 300 °C, as demonstrated in Figure 9. It is probable that the reduced ductility observed in the alloys extruded at lower temperatures resulted from the occurrence of twins within the non-DRXed coarse grains. The optical micrographs near the fractured surfaces after the tensile tests showed that the strong twin formation mostly appeared in the alloys extruded at 250 °C and 300 °C. It can be seen that the twins were preferentially formed within the coarse and elongated non-DRXed grains.

The SEM microstructures of the fractured surfaces are presented in Figure 10. The fracture surfaces of the as-cast and homogenized alloys consisted of mostly cleavages, tearing edges, and several large and shallow dimples. The second-phase particles were evidently broken during the tensile test and were responsible for the low ductility. However, the fracture surfaces of the extruded alloys mainly contained small dimples, in which cracked Mg–Zn–Gd ternary phase particles were mostly located at their center. The diameters of the dimples seemed to grow with increasing extrusion temperatures. Moreover, the alloy extruded at 250 °C also contained large cleavage planes, which represented the fracture of non-DRXed coarse grains.

### 3.3. Corrosion

Figure 11 depicts the weight loss per unit area over time for the as-cast, homogenized, and extruded alloys following their immersion in 3.5% NaCl for 48 h. It is evident that the homogenized alloy showed the poorest corrosion performance among the studied alloys with a mass corrosion rate of 0.32 mg cm^−2^ h^−1^. This indicated that the microstructural modification involving the partially dissolved second-phase particles and increased α-Mg grain size after the homogenization treatment played an important role in the corrosion of magnesium alloys [49,50]. The reduction in the amount of second-phase particles, which are cathodic to the α-Mg matrix, and decreased grain size were reported to improve the corrosion resistance of magnesium alloys [49,51]. As mentioned earlier, the homogenization treatment led to a dissolution of second-phase particles and an increase in the grain size. Thus, one might deduce that the negative impact of the larger grains outweighed the positive effect of the reduction in the amount of second-phase particles on the corrosion resistance. However, it should be considered that the significant alteration in the morphology of the secondary phases may also be very important for the initiation and progression of corrosion, which will be discussed later. After the extrusion process, the corrosion resistance of the GZX220 alloy was markedly increased. With an increase in extrusion temperature from 250 °C to 300 °C, the corrosion rate decreased by around 19%, above which it displayed negligible variations. Earlier research has also indicated that the corrosion resistance can be improved through an extrusion process [49,52,53]. The general consensus was that the enhanced corrosion resistance in the extruded alloys was primarily attributed to the creation of a more adherent passive layer, which resulted from an increased amount of high-angle grain boundaries.

SEM micrographs of the corroded surfaces of the alloys after 48 h of immersion in 3.5% NaCl without corrosion products are presented in Figure 12. The macroscopic images of each sample are also demonstrated in Figure 12. Localized corrosion attacks were observed to be severe in the as-cast and homogenized alloys in which the degree of corrosion localization was higher in the homogenized alloy having relatively large and deep pits. It is also clear that the corrosion mainly appeared as the dissolution of the matrix phase, and even the morphology of the second phases along the grain boundaries can be observed especially in the as-cast alloy, which can be seen at the high-magnification microstructure in Figure 12. The extrusion process seemed to provide much more uniform corrosion compared to the as-cast and homogenized states. The alloy extruded at 250 °C displayed the greatest corrosion propagation with few localized areas among the extruded samples. With the increase in the temperature up to 400 °C, corrosion took place much more homogenously, generating evenly distributed corrosion products throughout the surfaces with almost no pitting corrosion. Previous studies revealed that corrosion generally initiated in the direct vicinity of the Mg–Zn–RE ternary particles, followed by the dissolution of the α-Mg phase [49,54,55]. This was due to the strong micro-galvanic interaction between second-phase particles and α-Mg. Yin et al. [56] studied the volta potential differences between the W-phase (Mg_3_Zn_3_Gd_2_), I-phase (Mg_3_Zn_6_Gd), and X-phase (Mg_12_ZnGd) and the α-Mg matrix and reported the values of potential differences as 120 mV, 25 mV, and 60 mV, respectively. In the present study, a large number of W-phases was present in the microstructures, and thus, localized corrosion propagations in the as-cast and homogenized alloys were attributed to the strong micro-galvanic coupling effect of the W-phase and the α-Mg matrix phase, which expedited the corrosion of the alloys. The deteriorated corrosion performance after homogenization treatment can also be explained by the dual role of the cathodic second-phase particles depending on their morphologies. As shown in Figure 13a, cathodic second phases (primarily W-phase) in the as-cast alloy acted as corrosion initiation sites since they are more noble than the α-Mg matrix. Afterwards, the corrosion mainly occurred as the dissolution of the anodic matrix phase to the grain interiors. Eventually, some parts of the second phases detached or fell off after the exhaustion of α-Mg grains. However, it is clearly indicated in the cross-section micrograph of the as-cast alloy in Figure 13a that the second phases with partially connected network structures acted as an effective barrier and retarded the corrosion development by separating grains from each other. The barrier effects of some cathodic second phases against the corrosion of magnesium alloys were also reported in several studies [56,57,58]. On the contrary, this barrier effect must have ceased due to the partial dissolution of the second phases after homogenization treatment, which left the skeleton-like morphology of the second phases behind together with thin rods or discrete tiny particles. Thus, the abundant micro-galvanic coupling caused serious corrosion in the homogenized alloy with no barrier effect, which generated a severe weight loss, as shown in Figure 11. Furthermore, as shown in Figure 13b, uniform corrosion with no formation of pitting was observed in the alloy extruded at 350 °C exhibited. Since the extrusion process led to a microstructure with very fine DRXed grains and homogenously distributed fragmented second-phase particles, the oxide film formed during the corrosion remained more stable, diminishing the harmful effect of cathodic phases.

Figure 14a presents the potentiodynamic polarization curves for the as-cast, homogenized, and extruded alloys after immersion in 3.5% NaCl for 30 min at room temperature. The values of the cathodic corrosion current density (*i_corr_*) and the corrosion potential (*E_corr_*) derived from the polarization curves are also presented in Figure 14b. The *i_corr_* values were measured by Tafel extrapolation of the cathodic branch since magnesium alloys exhibit abnormal anodic behavior (negative difference effect) [57]. It is agreed that the cathodic branch of the polarization curve represents the hydrogen evolution, whereas the dissolution of the Mg matrix is represented in the anodic branch [14,59]. The cathodic branches of the alloys showed an extensive linear behavior with similar progression. This implies that the cathodic reactions did not show a considerable change with processing conditions and the hydrogen evolution reactions of the alloys were steady, resulting in uniform corrosion during the cathodic polarization [56]. As presented in Figure 14b, the cathodic current density showed a peak point for the homogenized alloy accompanied by the most negative corrosion potential. This demonstrates that the homogenization resulted in the worst corrosion resistance among the studied processing conditions, which is consistent with the results obtained from the immersion tests. Moreover, the studied samples did show either passivation region or clear breakdown potential in their anodic branches, indicating the formation of an incomplete protective oxide film on the surfaces. The increasing extrusion temperature caused lower cathodic corrosion current densities. As given in Figure 14b, both the *i_corr_* and *E_corr_* values showed a gradual decrease as the extrusion temperature increased from 250 °C to 400 °C. Since lower *i_corr_* and more positive *E_corr_* values indicate the lower corrosion rates and higher nobility of the metal, respectively, it can be concluded that the increasing extrusion temperatures continually improved the corrosion performance of the GZX220 alloy even though the cathodic current densities of the alloys extruded at temperatures above 300 °C were nearly the same. As outlined above, the enhancement of the corrosion performance after the extrusion processes can be attributed to the modification of the grain structure and cathodic W-phase [26,31,56]. The main corrosion in the homogenized alloy initiated around these particles and propagated through the grain interiors with no barrier effect, resulting in a significant dissolution and weight loss. After the extrusion process, the coarse W-phase was fragmented into fine particles and the severity of the corrosion propagation ceased. Furthermore, a refined DRXed grain structure was obtained with the extrusion process, which can also contribute to the formation of a more stable protective oxide layer. The increase in the extrusion temperature also led to a higher degree of DRX and a more uniform microstructure. Thus, the corrosion performance was further improved by the increasing extrusion temperature. Tafel results were found almost in line with the immersion corrosion test results, shown in Figure 11.

The EIS results of the as-cast, homogenized, and extruded alloys after immersion in 3.5% NaCl for 30 min at room temperature are depicted in Figure 15 in the form of Nyquist and Bode plots. Evidently, all the alloys exhibited similar spectra having one big high-frequency capacitive loop, followed by an ambiguous time constant in the lower-frequency regime. No inductive loop was observed in the present study. This is basically because the EIS tests were conducted at a lower limit frequency of 0.1 Hz, and below this limit, some non-stationarities are likely generated due to the fast dissolution of magnesium, resulting in an appearance of a pseudo-inductive loop [60,61]. Although there have been disagreements among the researchers about the interpretation of time constants in EIS spectra of magnesium alloys, recent studies by Wang et al. [62,63] have shed light on the origin of the capacitive loops in Nyquist plots, where the high-frequency capacitive loop results from the barrier properties of the surface film (oxide/hydroxide layer) and the middle-frequency capacitive loop is due to the charge transfer process and the respective double-layer structure at the electrode/electrolyte interface. As shown in Figure 15a, the arc diameters of the high-frequency capacitive loops noticeably varied from each other, suggesting that microstructural modifications by different processing conditions had an impact upon the properties of the surface film. The alloys extruded at temperatures of 300 °C and above had relatively larger high-frequency capacitive loops than the as-cast and homogenized alloys. Furthermore, the alloy extruded at 350 °C revealed by far the largest capacitive loop in the Nyquist plots in Figure 15a together with the highest total impedance at low frequencies and the highest phase angle in the Bode plots in Figure 15b among the studied alloys, indicating the best barrier properties of the surface film and corrosion resistance. The equivalent circuit model for the EIS curves is also presented in Figure 16, where *R_s_* is the solution resistance, *R_f_* is the corrosion product resistance, *R_ct_* is the charge transfer resistance, and *CPE_f_* and *CPE_dl_* are the constant phase elements referring to the capacitance of corrosion product layer and electric double layer, respectively. The parameters determined by fitting with the equivalent circuit are given in Table 4. The polarization resistance values, *R_p_*, are also presented in Table 4, which were calculated by the following equation on the basis of the infinite impedance of capacitive components as the frequency tends to zero:(1)$$R_{p}=R_{f}+R_{ct}$$

It is apparent that the *R_p_* values decreased after homogenization and gradually increased following extrusion processes up to the extrusion temperature of 350 °C, above which it slightly reduced. Since *R_p_* is inversely proportional to corrosion rate according to the Stern–Geary relationship [64], the higher *R_p_* values indicated lower corrosion rates due to the formation of a more protective layer of stable corrosion products. As stated above, the abundant micro-galvanic coupling of the cathodic W-phase and the anodic α-Mg matrix phase gave rise to a higher dissolution in the homogenized alloy, inducing a low polarization resistance. Moreover, the homogenized alloy exhibited the highest double-layer capacitance value (*CPE_dl_*), which stemmed from the larger corrosion area. The extrusion process with an increasing initial process temperature showed a tendency to cause a more stable protective oxide film due to the fine DRXed grains and uniformly distributed second-phase particles. It is also worth noting that the extrusion at temperatures as low as 250 °C did not yield a noticeable improvement in the corrosion resistance as the alloy extruded at 250 °C displayed a poor polarization resistance. This can be ascribed to the significant effect of the extrusion temperature on the preferential crystallographic orientation (see Figure 7), in which the alloy extruded at 250 °C showed a strong basal texture intensity due to the high fraction of non-recrystallized grain. It was reported that the bimodally distributed (i.e., partially DRXed) microstructures may show a galvanic coupling between the basal and prismatic planes resulting in a deterioration in the corrosion resistance [65,66]. Thus, the extruded alloys with a low degree of DRX and a strong texture showed somewhat underdeveloped corrosion properties.

## 4. Conclusions

Based on the findings, the following conclusions can be made:The as-cast alloy consisted mainly of α-Mg, Mg_3_Gd_2_Zn_3_ (also denoted as W-phase), and Ca_2_Mg_6_Zn_3_ phases. After the homogenization treatment, the α-Mg grain size increased, and the second-phase particles underwent partial dissolution.A substantial grain refinement was obtained by the extrusion process. Increasing the extrusion temperature led to a rise in the proportion of DRXed grains and a decline in their size. At low extrusion temperatures, higher basal texture intensities were observed.The mechanical properties exhibited a significant enhancement after the extrusion procedure. The alloy extruded at 250 °C showed the highest strength but low ductility, which improved up to the extrusion at 350 °C above which it decreased.The as-cast alloy displayed a moderate corrosion resistance having a barrier effect of secondary particles. However, after the homogenization treatment, the corrosion resistance deteriorated by the increased micro-galvanic coupling effect. The extrusion process with an increasing initial temperature of up to 350 °C resulted in a gradual improvement of the corrosion resistance.

## Figures and Tables

**Figure 1 materials-16-03075-f001:**
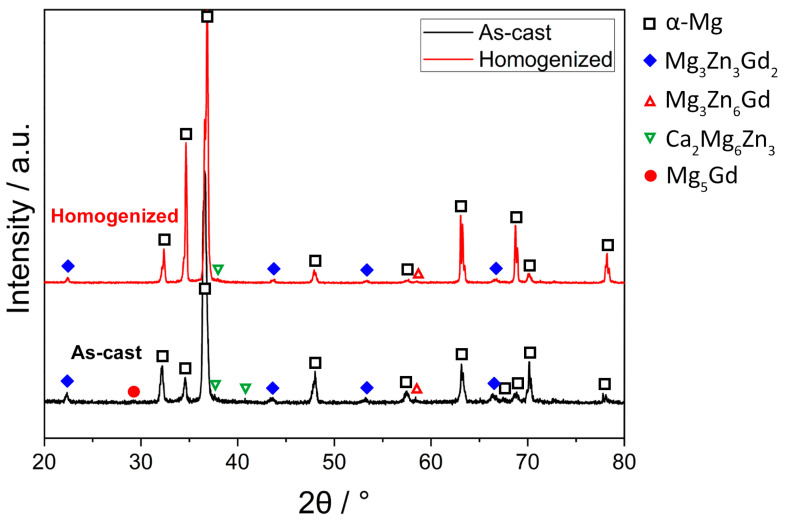
XRD spectra for the as-cast and homogenized alloys.

**Figure 2 materials-16-03075-f002:**
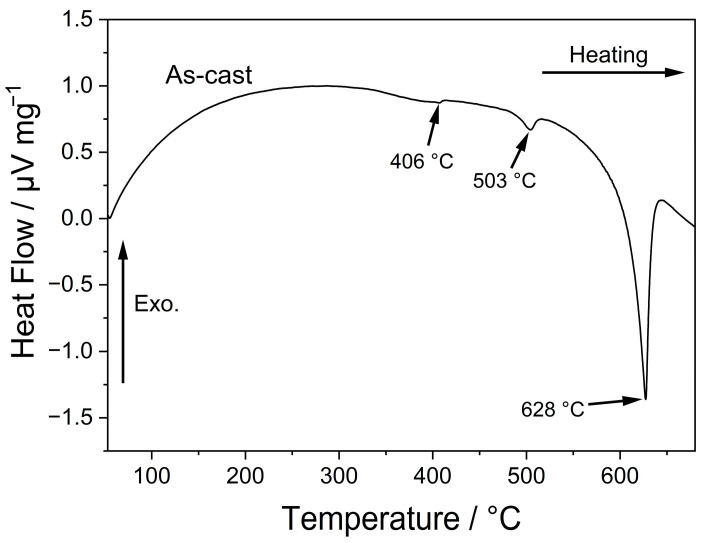
DSC result of the as-cast alloy.

**Figure 3 materials-16-03075-f003:**
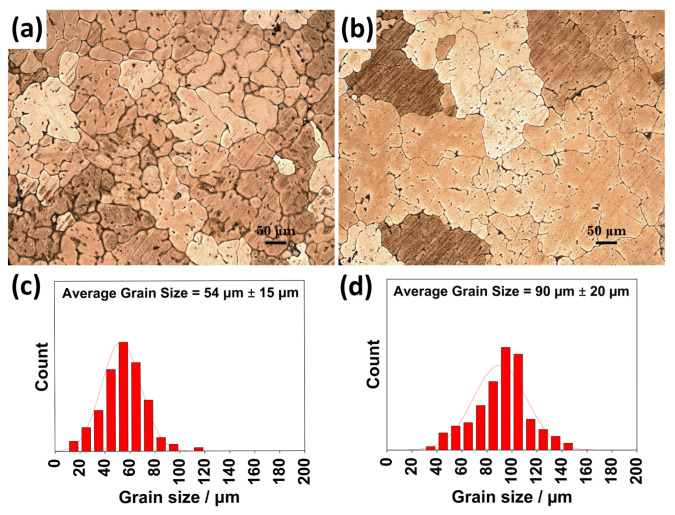
Optical micrographs and grain size distributions of (**a**,**c**) as-cast and (**b**,**d**) homogenized alloys.

**Figure 4 materials-16-03075-f004:**
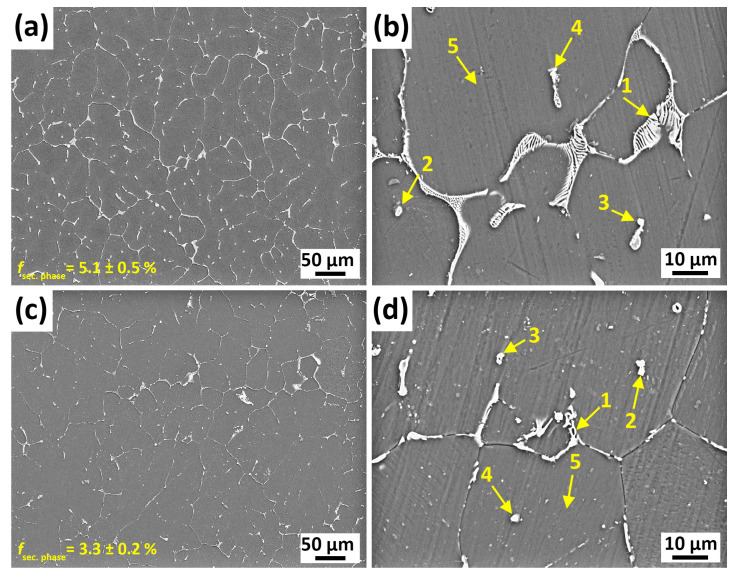
SEM micrographs of (**a**,**b**) as-cast and (**c**,**d**) homogenized alloys.

**Figure 5 materials-16-03075-f005:**
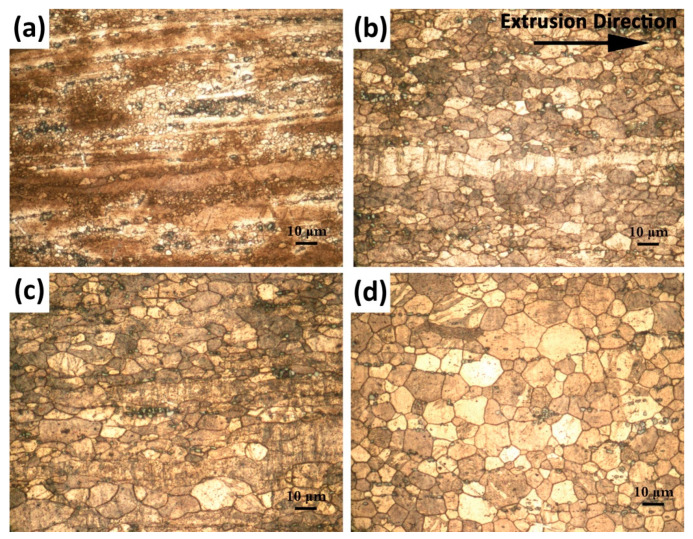
Optical micrographs of the alloys extruded at (**a**) 250 °C, (**b**) 300 °C, (**c**) 350 °C, and (**d**) 400 °C.

**Figure 6 materials-16-03075-f006:**
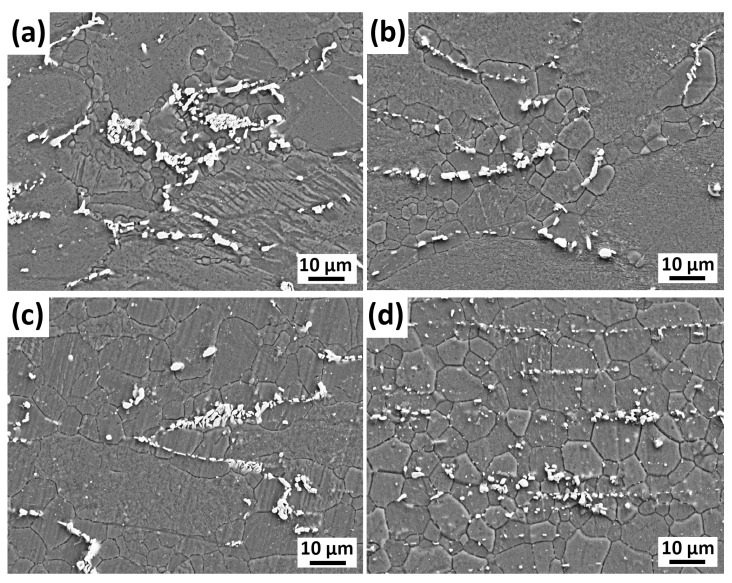
SEM micrographs of the alloys extruded at (**a**) 250 °C, (**b**) 300 °C, (**c**) 350 °C, and (**d**) 400 °C.

**Figure 7 materials-16-03075-f007:**
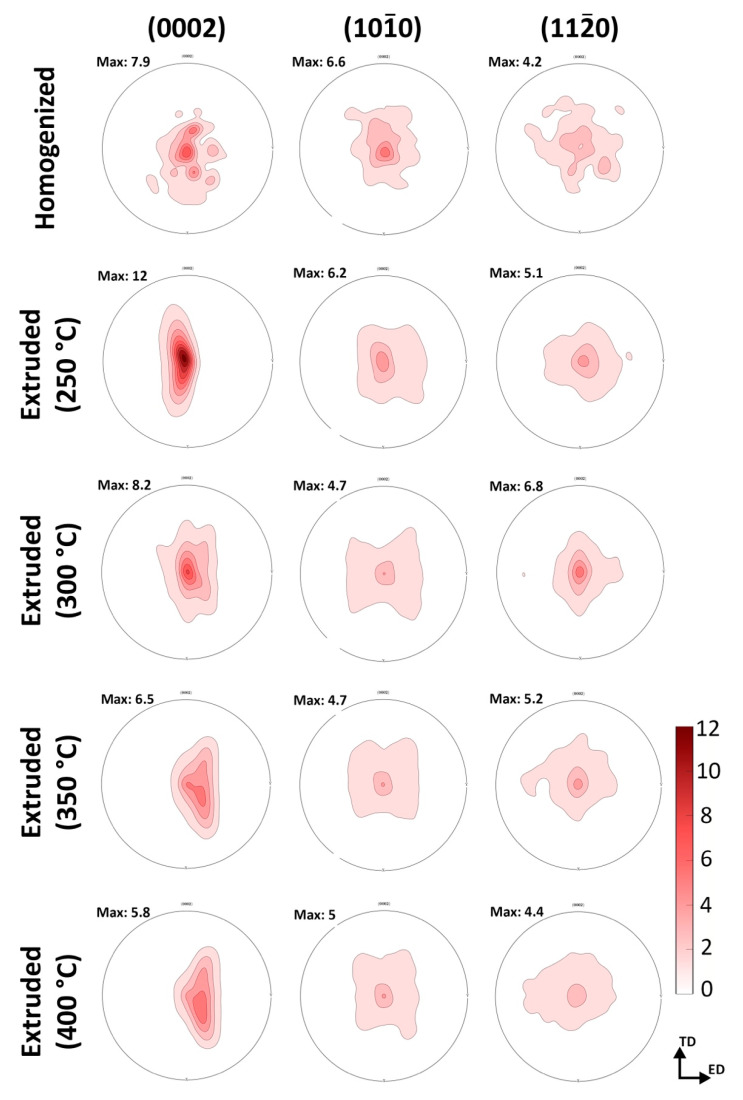
The (0002), (10$\bar{1}$0), and (11$\bar{2}$0) pole figures of the homogenized and extruded alloys.

**Figure 8 materials-16-03075-f008:**
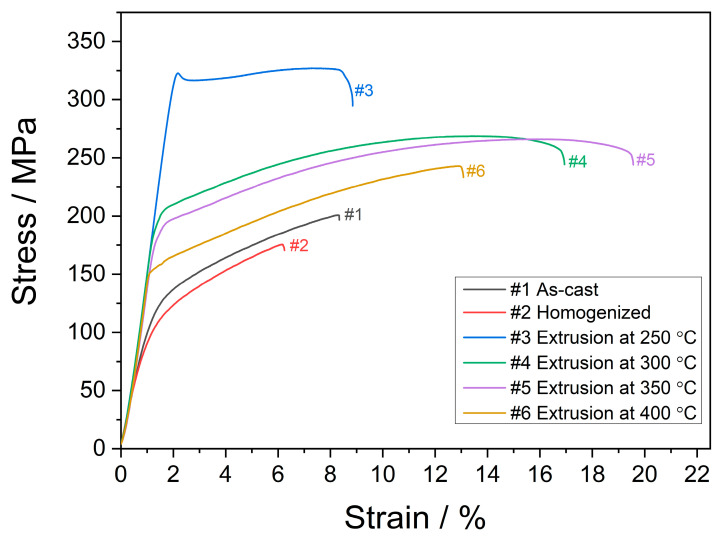
Engineering stress–strain curves for the as-cast, homogenized, and extruded alloys.

**Figure 9 materials-16-03075-f009:**
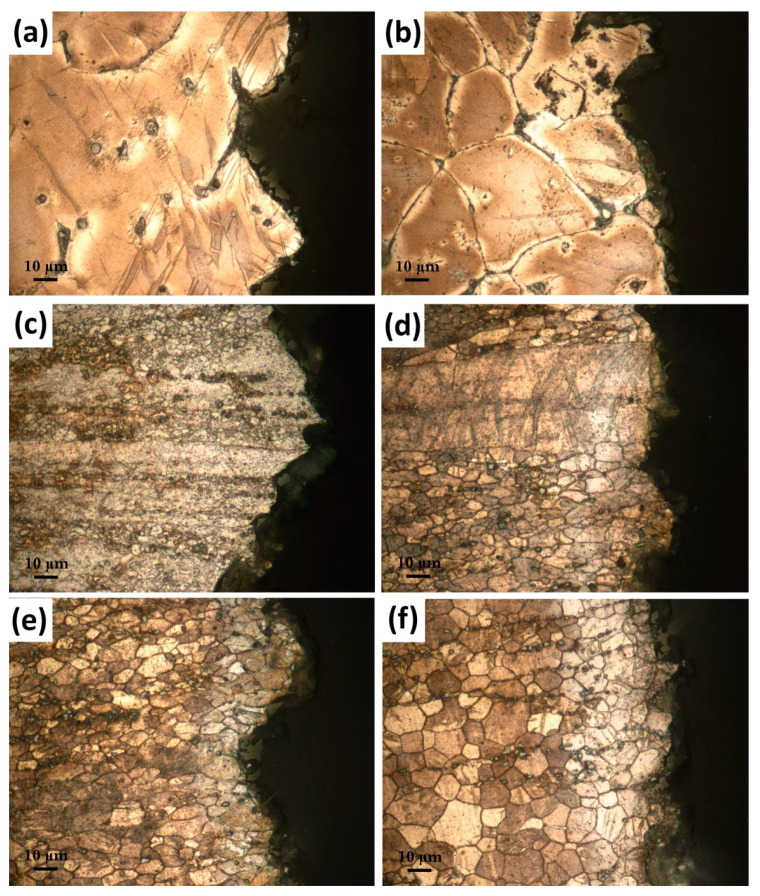
Cross-section optical micrographs of the fractured samples: (**a**) as-cast, (**b**) homogenized, extruded at (**c**) 250 °C, (**d**) 300 °C, (**e**) 350 °C, and (**f**) 400 °C.

**Figure 10 materials-16-03075-f010:**
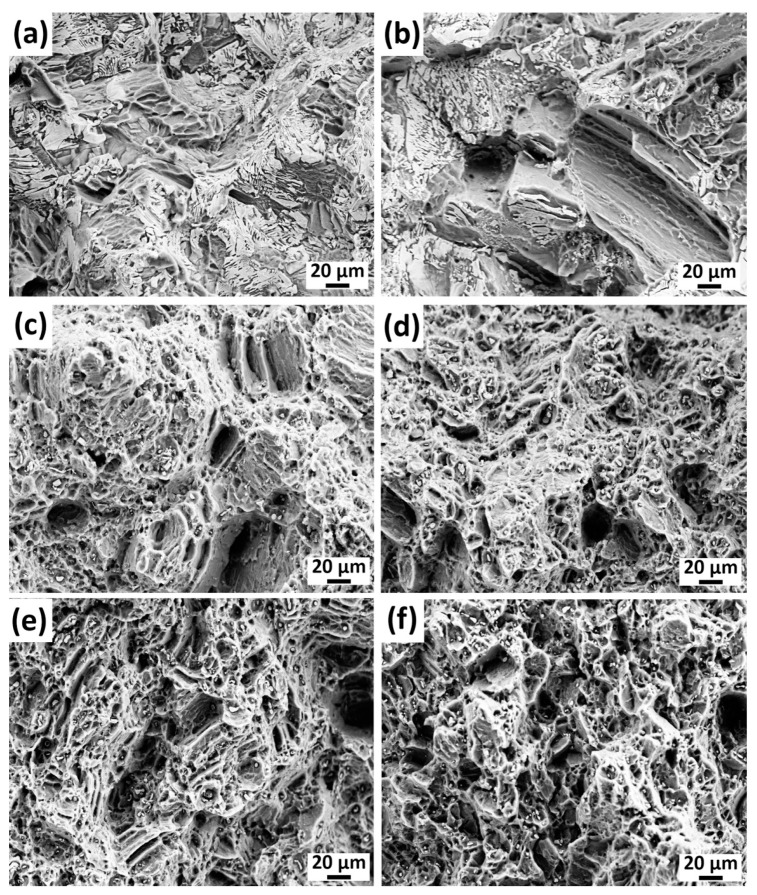
SEM micrographs of the fractured surfaces after tensile tests: (**a**) as-cast, (**b**) homogenized, extruded at (**c**) 250 °C, (**d**) 300 °C, (**e**) 350 °C, and (**f**) 400 °C.

**Figure 11 materials-16-03075-f011:**
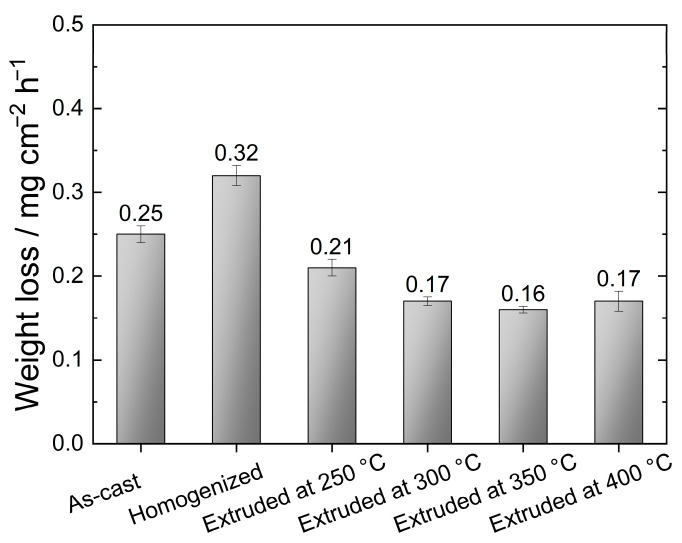
Weight loss of the alloys after 48 h of immersion in 3.5% NaCl solution.

**Figure 12 materials-16-03075-f012:**
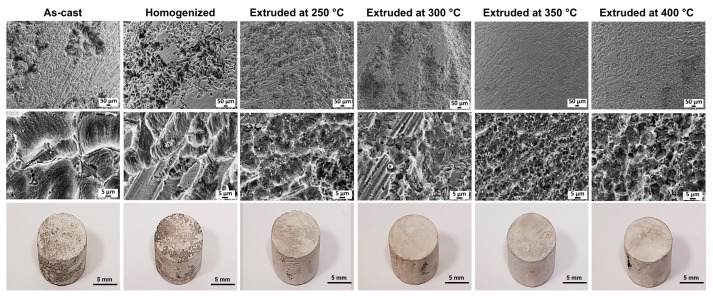
SEM micrographs of the corroded surfaces after 48 h of immersion in 3.5% NaCl with the macroscopic images of the cleaned samples.

**Figure 13 materials-16-03075-f013:**
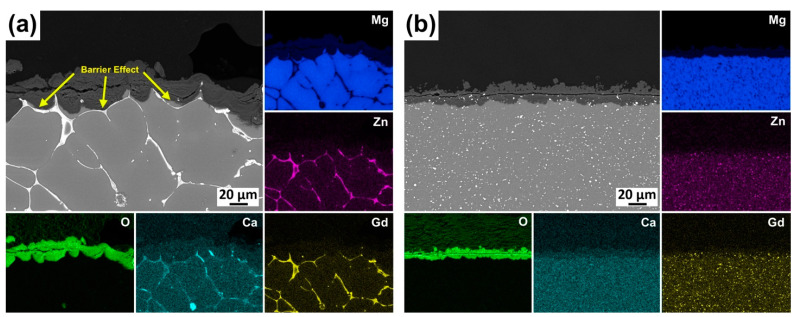
Cross-section SEM micrographs after 48 h of immersion in 3.5% NaCl with the associated EDS map of the elements in (**a**) the as-cast alloy and (**b**) the alloy extruded at 350 °C.

**Figure 14 materials-16-03075-f014:**
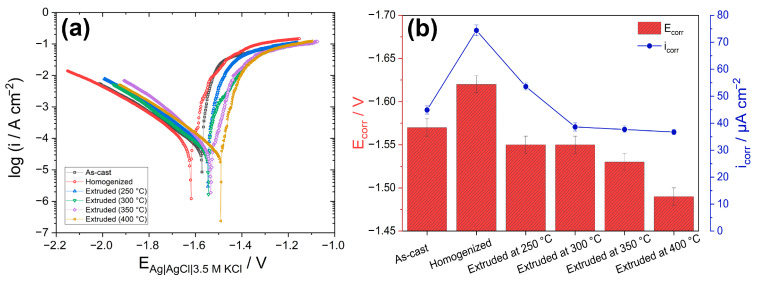
Polarization test results of the alloys: (**a**) potentiodynamic polarization curves and (**b**) *E_corr_* and *i_corr_* parameters derived from the polarization curves.

**Figure 15 materials-16-03075-f015:**
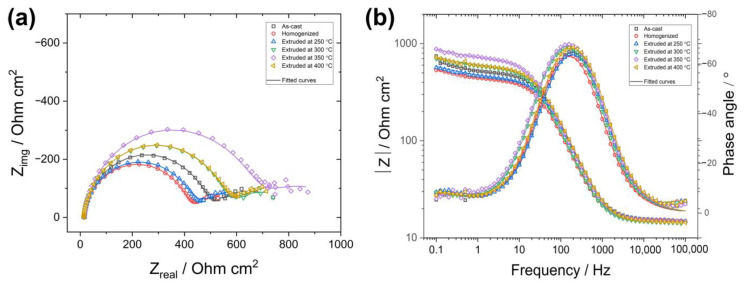
EIS curves: (**a**) Nyquist plots and (**b**) Bode plots.

**Figure 16 materials-16-03075-f016:**
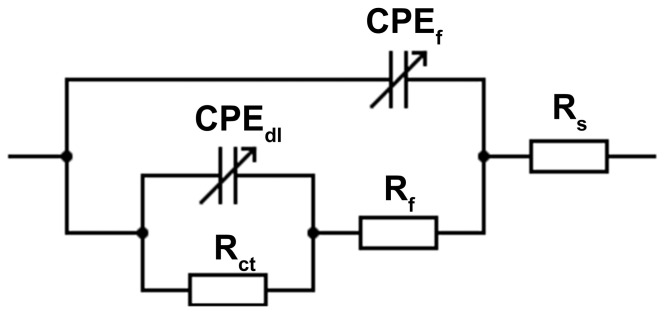
Equivalent electrical circuit to model EIS curves.

**Table 1 materials-16-03075-t001:** XRF results of the as-cast alloy.

Alloy	Compositions/wt%			
	Zn	Gd	Ca	Mg
GZX220	2.23	2.21	0.19	Bal.

**Table 2 materials-16-03075-t002:** EDS results of the points in Figure 4.

Condition	Spectrum	Elements							
		Mg		Zn		Gd		Ca	
		/wt%	/at%	/wt%	/at%	/wt%	/at%	/wt%	/at%
As-cast	1	15.5	36.6	55.6	48.7	24.6	8.9	4.2	5.7
	2	22.6	46.5	50.0	38.3	20.4	6.5	7.0	8.7
	3	30.6	51.8	50.1	31.6	4.2	1.1	15.1	15.5
	4	42.8	69.5	39.7	24.1	14.3	3.6	2.9	2.6
	5	96.2	98.6	2.4	0.9	0.8	0.1	0.5	0.3
Homogenized	1	20.1	44.5	56.6	46.6	22.3	7.7	0.8	1.1
	2	32.7	69.7	6.3	5	53.2	17.5	4.6	6.1
	3	14.2	35.6	57.7	53.4	27.8	10.6	0.3	0.5
	4	11.3	29.2	62.3	59.9	26.1	10.4	0.3	0.5
	5	94.9	98.2	3.1	1.2	1.3	0.2	0.4	0.3

**Table 3 materials-16-03075-t003:** Tensile test results.

Alloy	0.2% YS/MPa	UTS/MPa	Strain at Failure/%
As-cast	114 ± 6	200 ± 7	8.3 ± 1.4
Homogenized	95 ± 5	176 ± 6	6.2 ± 1.8
Extruded (250 °C)	321 ± 4	326 ± 4	8.8 ± 1.1
Extruded (300 °C)	197 ± 4	269 ± 7	16.9 ± 1.8
Extruded (350 °C)	176 ± 5	266 ± 7	19.5 ± 2.1
Extruded (400 °C)	155 ± 6	242 ± 8	13.1 ± 1.4

YS: yield strength, UTS: ultimate tensile strength.

**Table 4 materials-16-03075-t004:** Results obtained from the EIS data fitting.

Specimen	*R_s_* /Ω cm^2^	*R_f_* /Ω cm^2^	*R_ct_* /Ω cm^2^	*CPE_f_* 10^−5^ /Ω^−1^ s^n^ cm^−2^	*n* _1_	*CPE_dl_* 10^−3^ /Ω^−1^ s^n^ cm^−2^	*n* _2_	*R_p_* /Ω cm^2^
As-cast	15.2	470.7	372.7	1.59	0.94	5.35	0.58	843.4
Homogenized	15.3	397.9	321.7	1.62	0.94	6.17	0.57	719.5
Ext. (250°)	15.1	419.7	360.9	1.69	0.94	5.83	0.61	780.6
Ext. (300°)	14.5	545.5	285.4	1.55	0.94	4.10	0.67	830.9
Ext. (350°)	14.9	654.5	430.2	1.69	0.93	3.63	0.54	1084.7
Ext. (400°)	14.6	542.0	447.9	1.53	0.94	4.82	0.57	989.9

## Data Availability

All the data is available in this manuscript.

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
