# Peer review of "Evolution of Microstructure, Mechanical Properties, and Corrosion Resistance of Mg–2.2Gd–2.2Zn–0.2Ca (wt%) Alloy by Extrusion at Various Temperatures"

_materials, 2023, doi:10.3390/ma16083075_

Round 1
Reviewer 1 Report
This nice submission needs a few corrections:
There are some English mistakes.
The caption of the macroscopic images shown in Fig. 12, as stated in the text, too, is located at Fig. 11.
Author Response
Respond to Reviewer 1:
- There are some English mistakes.
Respond: The quality of the English language was double-checked by several native speakers and related corrections were made in the manuscript.
- The caption of the macroscopic images shown in Fig. 12, as stated in the text, too, is located at Fig. 11.
Respond: Thank you for this comment. The corresponding figures were corrected.
Reviewer 2 Report
The authors study the evolution of microstructure, mechanical properties and corrosion resistance of Mg–2.2Gd–2.2Zn–0.2Ca (wt%) alloy after extrusion at various temperatures. They only provide results in terms of microstructure, mechanical properties and corrosion resistance, respectively. However, they don't discuss the relationships among microstructure, mechanical properties and corrosion resistance. Thus, we cannot understand the scientific issue and importance of this study.
Author Response
- The authors study the evolution of microstructure, mechanical properties and corrosion resistance of Mg–2.2Gd–2.2Zn–0.2Ca (wt%) alloy after extrusion at various temperatures. They only provide results in terms of microstructure, mechanical properties and corrosion resistance, respectively. However, they don't discuss the relationships among microstructure, mechanical properties and corrosion resistance. Thus, we cannot understand the scientific issue and importance of this study.
Respond: We appreciate your insightful feedback. As you pointed out, this research article presents findings on the microstructure, mechanical properties, and corrosion resistance of the magnesium alloy under different processing conditions. The main objective was to compare the mechanical and corrosion properties, with a sound understanding of the microstructural features. We believe that we have successfully established the correlations between the microstructure/mechanical properties and microstructure/corrosion resistance in this study. It is worth mentioning that mechanical properties and corrosion resistance are two separate fields that require independent investigation, unless mechano-corrosion mechanism such as erosion-corrosion, stress corrosion cracking, or corrosion fatigue are considered. Thus, no direct correlation between the mechanical and corrosion properties was made in this work, but the discussions were grounded on the same microstructural analysis. The significance of this study lies in the development of a new magnesium alloy and understanding of the effects of processing conditions that can offer superior corrosion resistance, strength, and ductility. We hope that our findings will contribute to the advancement of knowledge on this promising alloy system.
Reviewer 3 Report
Notes on the article of Hüseyin Zengin, Soner Ari, Muhammet Emre Turan and Achim Walter Hassel «Evolution of microstructure, mechanical properties and corrosion resistance of Mg–2.2Gd–2.2Zn–0.2Ca (wt%) alloy by extrusion at various temperatures»
The paper reports about the effect of different processing treatment on microstructure, mechanical and corrosion properties of the Mg-2Gd-2Zn-0.2Ca (wt.%) alloy. The authors showed that an increase in extrusion temperature leads to decrease in strength of the alloy. The best combination of ductility and resistance to chemical corrosion was received after extrusion at 350 °C. The article has a big practical importance. It is an interesting and well-written report, which should be published after revisions that are listed below:
1. How did the authors determine open current potential? The details should be added in Section 2.
2. The authors write: «Considering that Ca2Mg6Zn3 phase displayed a significant amount of dissolution during homogenization at 400 °C, it can be deduced that the transition peak at 406 °C corresponded to Ca2Mg6Zn3 phase. It can also be deduced from Fig.2 that the predominating W-phase exhibited a transition at 503 °C and the melting point of the alloy was 628 °C». However, four types of intermetallic compounds were discovered by XRD analysis (Fig. 1). Why peaks for other two compounds are absent?
3. The authors should use the same style of writing the stoichiometric composition of particles in the text and in Fig. 1. This is not a critical remark, but it may confuse readers a little.
4. The values of grain size with experimental errors were calculated by the authors (Fig. 3). However, the authors use the values without experimental errors in text. It should be used the first version in both cases.
5. Figure 8 shows the stress – strain curves for the alloy in different states. How can the authors explain the fact that a visible sharp yield point exists only for the 3rd curve? Is the reason in microstructure difference? Or in particles configuration?
6. The authors wrote that «Each test was repeated at least three times». That’s why, the experimental errors values should be added to mechanical characteristics values.
Author Response
- How did the authors determine open current potential? The details should be added in Section
Respond: Thank you for the comment. The details about the OCP measurements were added to the Section 2 as follows:
“The electrochemical corrosion tests were performed by CompactStat potentiostat (Ivium Technologies, The Netherlands) after immersing in the chloride solution and monitoring the open circuit potential values for 30 min.”
- The authors write: «Considering that Ca2Mg6Zn3phase displayed a significant amount of dissolution during homogenization at 400 °C, it can be deduced that the transition peak at 406 °C corresponded to Ca2Mg6Zn3 It can also be deduced from Fig.2 that the predominating W-phase exhibited a transition at 503 °C and the melting point of the alloy was 628 °C». However, four types of intermetallic compounds were discovered by XRD analysis (Fig. 1). Why peaks for other two compounds are absent?
Respond: This was mainly due to the low fractions of the other two phases that did not gie any visible transition in the DSC peaks. The following statement was added to the manuscript:
“On the other hand, the DSC results did not reveal any peaks associated with the I-phase and Mg5Gd, as their fractions were expected to be very low according to the XRD results, which showed very weak intensities for these phases.”
- The authors should use the same style of writing the stoichiometric composition of particles in the text and in Fig. 1. This is not a critical remark, but it may confuse readers a little.
Respond: Thank you for the remark. The style of writing the compositions were kept same in the text but changed in the Fig. 1.
- The values of grain size with experimental errors were calculated by the authors (Fig. 3). However, the authors use the values without experimental errors in text. It should be used the first version in both cases.
Respond: Thank you, the experimental errors were added in the text as well.
- Figure 8 shows the stress – strain curves for the alloy in different states. How can the authors explain the fact that a visible sharp yield point exists only for the 3rdcurve? Is the reason in microstructure difference? Or in particles configuration?
Respond: Thank you for the comment. This is called yield point phenomenon which can be seen quite frequently in several metals such as Fe, Al, Ti etc. Even though it is not very common in Mg alloys, it might still be seen in some Mg alloys or after some processes. It is generally attributed to the locking of dislocations by solute atoms causing an increase in the initial yield stress (upper yield point) and the liberation of these dislocations which results in slipping at lower stress values (lower yield point). Since the present study focused only one type of alloy composition and this was only observed in the alloy extruded at the lowest temperature, this behaviour was clearly because of the degree of DRX. Because low extrusion temperature did not cause a full DRX with very few localized fine DRX grains and significant deformed non-DRXed coarse grains. These deformed grains had already contained high amount of stress caused by high dislocation density. On the other hand, the DRX resulted in newly nucleated and so called ‘dislocation-free’ grains. Therefore, we thought that this bimodal structure was likely the main reason of this behavior in which the lock of dislocation in the non-DRXed grains was liberated at some stress values by DRXed grains and slip occurred in balance at lower stresses. We can humbly make such deductions here as an answer to your comment since this needs to be further investigated or it can be even a subject of another study. But still the determination of yield and ultimate tensile points can be made reliably in this figure.
- The authors wrote that «Each test was repeated at least three times». That’s why, the experimental errors values should be added to mechanical characteristics values.
Respond: The experimental errors were added to the relevant parts in the manuscript.
Reviewer 4 Report
Manuscript ID: materials-2327199
INTRODUCTORY COMMENTS
In this manuscript, the Mg-2.2Gd-2.2Zn-0.2Ca (wt%) alloy (GZX220) was cast by permanent mold casting, followed by homogenization at 400 °C for 24 h and extrusion at four different temperatures: 250 °C, 300 °C, 350 °C and 400 °C. Microstructural analyses showed that the as-cast alloy consisted of α-Mg, Mg-Gd binary and Mg-Gd-Zn ternary phases. Besides, the extrusion process resulted in a remarkable improvement of the mechanical properties. It's a valuable experimental work, with good organization and language.
I think this work is interesting to scientific community, especially for advanced magnesium alloys scientists and related experiment researchers.
However, a minor revision is needed to allow the readers to appreciate the work better. Please consider the comments below.
COMMENTS
1. In the introduction section, the content of the third paragraph can be richer, and there are many other scientists' work that can be summarized. Such as the electronic origin of the ductilizing of Mg with different kinds of solutes, including Gd, Zn and Ca.
2. Could you explain what are the main different between Gd and Ca for ductilizing the magnesium alloy? For example, Gd can decrease the activation of <c+a> dislocation and the formation of twinning, thus benefit the ductility of Mg. Dose Ca follow any other way to increase the ductility and strength of Mg?
3. In figure 4 and 6, there are some white precipitates, what are they? Is it possible to form LPSO?
4. In this work, it seems that the extrusion temperature of 400 degrees is not as good as 350 degrees. Can you provide a microscopic explanation?
5. If possible to do some TEM, and find out the relationship between microstructure, especially precipitation phase and grain boundary segregation, and mechanical properties.
Author Response
Respond to Reviewer 4:
- In the introduction section, the content of the third paragraph can be richer, and there are many other scientists' work that can be summarized. Such as the electronic origin of the ductilizing of Mg with different kinds of solutes, including Gd, Zn and Ca.
Respond: Thank you for your valuable comment. The introduction part has been revised including ductilizing effects used elements. The following part was added to the Introduction Section. This also involves answers to Comment 2.
“A recent research by Gu et al. [20] demonstrated that adding up to 4 wt% of Gd to magnesium effectively enhanced its basal slip resistance compared to prismatic slip resistance, ultimately leading to an increase in the ductility of the magnesium. Furthermore, another study suggested that the presence of Gd, particularly when combined with Zn, may influence the texture development of magnesium and subsequently enhance its mechanical strength by increased bonding of solute elements to grain boundaries [21]. A parameter to be considered while designing magnesium alloys is the "electronic work function (EWF)", which provides insight into the behavior of electrons in the metal. According to Liu and Li [22], if the EWF value of the dissolved element is smaller than that of Mg (3.66 eV), both strength and ductility can increase. Strength increase occurs with solid-solution strengthening. The ductilizing effect is due to the dissolved element creating cross-slip stress softening (a-type dislocations) that provides dynamic recovery during deformation, and new nucleation zones for pyramidal <c+a> dislocations as a result of lower I1 stacking fault energy. Zn, with an EWF value of 4.33 eV, can effectively enhance the strength of magnesium through solid solution strengthening. However, Gd has an EWF value of 3.17 eV and is an effective alloying element in increasing both the strength and ductility of magnesium. Another element that can have a similar effect is calcium (Ca), which is frequently used in magnesium alloys with an EWF of 2.87 eV. It was reported that alloying Mg-Zn alloy with Ca can give rise to a decline in the intensity of the basal texture, a grain size reduction, and a substantial enhancement in the tensile elongation [23]. It was also showed that the minor Ca addition to a Mg-Zn-RE-Zr alloy reduced the yield asymmetry owing to the smaller grains and weaker texture [24].
- Could you explain what are the main different between Gd and Ca for ductilizing the magnesium alloy? For example, Gd can decrease the activation of <c+a> dislocation and the formation of twinning, thus benefit the ductility of Mg. Dose Ca follow any other way to increase the ductility and strength of Mg?
Respond: The related discussion was added to the Introduction section. Please see the added paragraph in the Respond 1 above.
- In figure 4 and 6, there are some white precipitates, what are they? Is it possible to form LPSO?
Respond: It is true that Mg-Zn-Gd systems can form LPSO phase at specific Zn/Gd ratios, especially when this ratio is lower than 1. In this study the atomic ratio of Zn to Gd is around 2.4, therefore it was not expected to have an LPSO phase in this alloy. Though no evidence was found regarding LPSO phase during microstructure characterizations. The white precipitates in Figures 4 and 6 corresponded to mainly W-phase (Mg3Zn3Gd2) and Ca2Mg6Zn3 ternary phases as confirmed by the XRD and EDS analyses.
- In this work, it seems that the extrusion temperature of 400 degrees is not as good as 350 degrees. Can you provide a microscopic explanation?
Respond: Regarding the corrosion resistance the extrusion temperatures of 350 and 400 °C extremely similar corrosion rates. Even the corrosion morphologies in Fig. 12 are quite comparable. However, there is a difference in terms of mechanical properties. This is basically due to the excessive grain coarsening and texture softening that observed in the alloy extruded at 400 °C. After the extrusion at 350 °C, the degree of DRX was also almost 100 % but the grain size is finer. This is why this alloy exhibited the optimal properties.
- If possible to do some TEM, and find out the relationship between microstructure, especially precipitation phase and grain boundary segregation, and mechanical properties.
Respond: We regret to say that we could not add the TEM analyses to this revision. We have contacted with several institutions but unfortunately they are either out of work or give an appointment for a very late date. Considering the duration of revision which is 10 days for Materials, it is not possible for us to perform TEM analysis in the given period. Thank you in advance for your understanding.
Round 2
Reviewer 2 Report
The authors have improved the introduction on the reason why they add Zn, Gd or Ca to magnesium alloys. In addition, they include more information as reviewers' suggestions.
1. However, in experiment part, they don't describe the detained operation parameters of pole figure measurement, 2 theta' angles of {0002}, {101Ì…0} and {112Ì…0} planes, Cu Ka radiation, an azimuth tilt range of 0–85o.
2. They just provide incomplete pole figures from 0 to 85 degrees. Thus, we cannot observer the position from 85 to 90 degrees. It is necessary to use software to recalculate and normalize the incomplete pole figures so that we can get the complete pole figure from 0 to 90 degrees.
Author Response
Dear Reviewer,
Thank you very much for your valuable comments. I hope the revisions we made would be satisfactory.
Kind regards,
Dr. Hüseyin Zengin
- However, in experiment part, they don't describe the detained operation parameters of pole figure measurement, 2 theta' angles of {0002}, {101Ì…0} and {112Ì…0} planes, Cu Ka radiation, an azimuth tilt range of 0–85o.
Respond: More information about the pole figure measurements were added to the Experimental Section as follow:
The experimental (0002), (10 0) and (11 0) pole figures were also measured by XRD method using Cu Kα radiation. The data was obtained on 5° tilt steps from 15° to 90° and azimuthal rotations over the entire 360° range.
2. They just provide incomplete pole figures from 0 to 85 degrees. Thus, we cannot observer the position from 85 to 90 degrees. It is necessary to use software to recalculate and normalize the incomplete pole figures so that we can get the complete pole figure from 0 to 90 degrees.
Respond: The pole figures were recalculated after normalization. Since we wanted to represent the pole figures on fixed scale (12) for a better benchmark of the different conditions and the plotting style was 'contourf' mode, the results might seem as incomplete pole figures. However, if it is required we can change the plotting style and give the each pole figure with their specific max. range. Indeed we tried both style and this looked better for the comparison. Thanks.
Reviewer 3 Report
The authors answered all question in details. The article can be accepted in the present form.
Author Response
Thank you.